# Progressive approach to eruption at Campi Flegrei caldera in southern Italy

Christopher R.J. Kilburn[1], Giuseppe De Natale[2] & Stefano Carlino[2]

Unrest at large calderas rarely ends in eruption, encouraging vulnerable communities to perceive emergency warnings of volcanic activity as false alarms. A classic example is the Campi Flegrei caldera in southern Italy, where three episodes of major uplift since 1950 have raised its central district by about 3 m without an eruption. Individual episodes have conventionally been treated as independent events, so that only data from an ongoing episode are considered pertinent to evaluating eruptive potential. An implicit assumption is that the crust relaxes accumulated stress after each episode. Here we apply a new model of elastic-brittle failure to test the alternative view that successive episodes promote a long-term accumulation of stress in the crust. The results provide the first quantitative evidence that Campi Flegrei is evolving towards conditions more favourable to eruption and identify field tests for predictions on how the caldera will behave during future unrest.

[1] UCL Hazard Centre, Department of Earth Sciences, UCL, Gower Street, London WC1E 6BT, UK. [2] INGV-Osservatorio Vesuviano, Via Diocleziano 328, Napoli 80124, Italy. Correspondence and requests for materials should be addressed to C.R.J.K. (email: c.kilburn@ucl.ac.uk).

Large calderas with areas of 100 km² or more are among the most-populated active volcanoes on Earth. They commonly show episodes of unrest at intervals of ∼10–10² years[1] and, although the minority end in eruption, each raises concern that volcanic activity might be imminent. An outstanding goal therefore remains to distinguish between pre-eruptive and non-eruptive episodes.

With an unprecedented 2,000-year record of historical unrest and eruption[2], Campi Flegrei provides key insights for understanding the dynamic evolution of large calderas. Three episodes of major unrest have occurred since 1950, in April 1950–May 1952, July 1969–July 1972 and June 1982–December 1984 (refs 3–5). The last occasion of such behaviour occurred during the century before the caldera's only historical eruption in 1538 (refs 2,6). The current unrest is consistent with a reactivation of the magmatic system after 412 years and, hence, with an increase in the threat from volcanic activity to the caldera's population of almost 360,000 people, as well as to the three million residents of Naples immediately outside its eastern margin.

The largest ground movements recorded since Roman times have been concentrated near the modern coastal town of Pozzuoli at the centre of the caldera (Fig. 1). They have been dominated by a secular subsidence of c. 1.7 m a century[2] that has been interrupted by at least two extended intervals of net uplift, by about 17 m in c. 1430–1538 (ref. 2) and about 3 m since 1950 (refs 5,7). The pattern of recent uplifts has been radially symmetric, decaying to negligible movements at distances of about 5 km from the centre in Pozzuoli[3,4,8]. The cause of deep-seated subsidence has to be confirmed, but the uplift is consistent with an elastic-brittle crust being pressurized at depths of about 2.5–3 km, near the base of the geothermal system (Fig. 1). Pressurization has been attributed to intrusions of magma, fed from a primary magma reservoir 7–9 km below the surface, and to disturbances of the geothermal system[8–16]. A sill geometry is preferred for the magma intrusions, because it requires the least overpressure to drive the observed magnitudes of uplift[15], and inversions of geodetic data for the 1970–1972 and 1982–1984 uplifts yield intruded volumes of 0.02–0.04 km³, sill diameters of 4–6 km and mean thicknesses on the order of metres (Fig. 1)[15,17].

Some 26,000 micro-earthquakes, or volcano-tectonic (VT) events, have been recorded across the central zone of the caldera during the current unrest (Fig. 1), about 80% of which have been located at depths between 1 and 3 km, and <3% at depths of 4 km or more[3,18–21]. More than 98% have had magnitudes of 2.5 or less[18], indicating the predominance of slip along faults ∼0.01–0.1 km across, or ten to a hundred times smaller than the dimensions of the deforming crust. The crust therefore contains a distributed population of faults that are much smaller than the dimensions over which deformation has occurred.

To evaluate the potential for eruption, conventional studies have focussed on interpreting the major unrest of 1982–1984 (refs 4,8–16). Implicit assumptions have been that the next unrest will resemble its predecessor and, hence, that the shallow crust and magmatic system at Campi Flegrei has returned to conditions similar to those before 1982. A necessary implication is that the potential for eruption will also be similar to that during 1982–1984. However, recent measurements from a pilot borehole for the Campi Flegrei Deep Drilling Project suggest that stress has instead been accumulating in the crust[22]. Successive episodes of uplift may thus be driving the crust towards a critical stress for bulk failure and, hence, to a greater potential for eruption than previously assumed.

We here propose that the whole sequence of unrest since 1950 belongs to a single, long-term evolutionary sequence of accumulating stress and crustal damage. We apply a new model

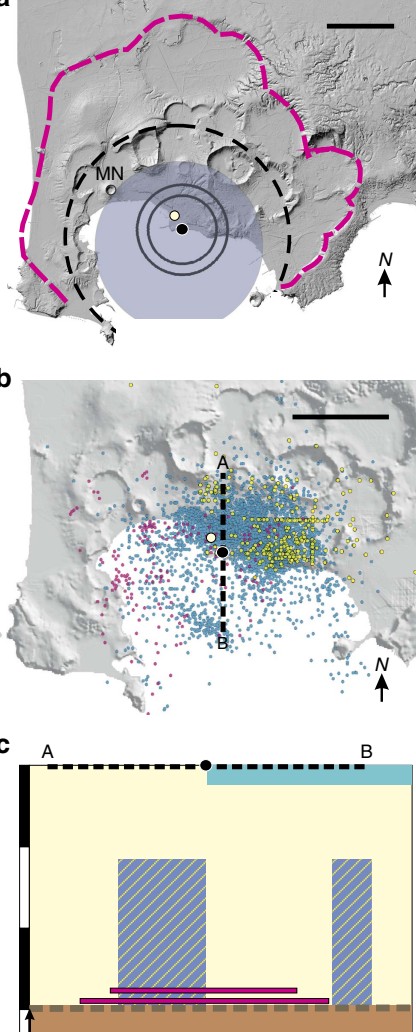

**Figure 1 | Structure and VT seismicity in Campi Flegrei. (a)** The active ring-fault zone encloses the principal area of caldera collapse (blue, shaded) associated with the eruption of the Neapolitan Yellow Tuff 15.6 kyr ago. Upward ground movements since 1969 have extended 5 km (black, dashed curve) around the centre of uplift (black filled circle) and within the perimeter (magenta, dashed curve) defined by the proposed collapse associated with the eruption of the Campanian Ignimbrite 39 kyr ago[54–56]. They are consistent with the intrusion of sills at depths of c. 3 km during 1969–1972 (inner black circle) and 1982–1984 (outer black circle). The Serapeo (white filled circle) in Pozzuoli is shown for reference. The scale bar is 3 km. **(b)** Epicentres of VT events[57] across Campi Flegrei for the intervals 1972–1974 (pink circles), 1983–1984 (blue circles) and 2005–2015 (yellow circles). The dashed line A–B shows the trace of the cross-section in **c**. The scale bar is 3 km. **(c)** Schematic north–south cross-section passing through the centre of uplift. The upper crust (pale yellow) that contains the hydrothermal system is underlain by bedrock (brown) beneath a thermo-metamorphic horizon (dashed brown line) at a depth of c. 3 km (ref. 56). Magmatic sills (magenta) were intruded at depths of c. 2.5 and 2.75 km during the 1969–1972 and 1982–1984 emergencies, respectively[17]. VT seismicity in 1982–1984 was concentrated inland from Pozzuoli and c. 2.5 km offshore[54], near the southern periphery of the inferred 1982–1984 sill (blue shading with yellow lines). The sea is shown in turquoise. The vertical thicknesses of the sills and sea are not to scale. The arrows show lengths of 1 km.

of elastic-brittle rock behaviour[23,24] to demonstrate that the increasing levels of VT seismicity associated with successive uplifts reflect changes in how the crust accommodates the strain energy supplied by magmatic intrusions. In particular, the behaviour follows the trend expected as the dominant factor controlling deformation changes from the elastic storage of strain energy to the release of that energy by faulting. Continuation of the trend will favour bulk failure in the crust and, hence, a greater potential for eruption than during previous emergencies. The results emphasize the importance of incorporating rock-physics criteria into strategies for evaluating the potential for eruption, especially at volcanoes that have yet to establish an open pathway for magma to reach the surface. They also highlight the need to raise awareness among vulnerable communities that a lack of eruption during recent emergencies cannot be used to infer that an eruption is also unlikely during a future crisis.

## Results

**Unifying episodes of unrest.** After correction for secular sub-sidence[2,15], the three major unrests at Campi Flegrei since 1950 have been characterized by initial uplifts for 2–3 years at mean rates of 0.3–0.6 m per year at the Serapeo in Pozzuoli, followed by minor corrected subsidence and subsequent recovery over 10–33 years (Fig. 2). The total corrected uplift at Serapeo has been c. 4 m (Fig. 2).

Rapid uplift occurs when the crust is extended over a newly intruded sill. We thus view the post-1950 unrest as equivalent to a total of 6–7 years of rapid uplift under increasing differential

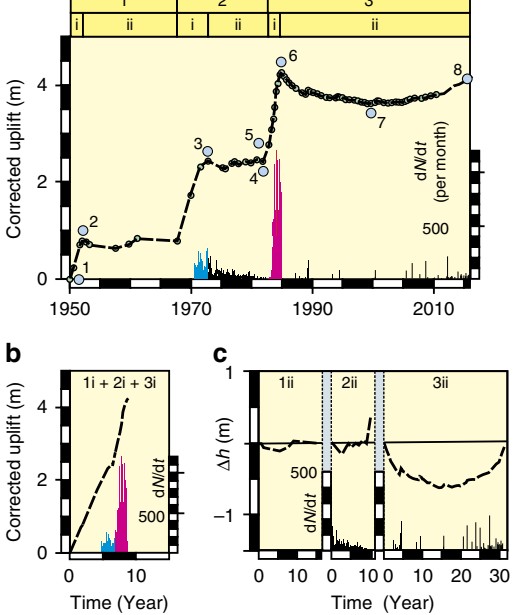

**Figure 2 | Uplift and VT events during unrest at Campi Flegrei. (a)** After a secular subsidence at 0.017 m per year has been removed, vertical ground movement at the Serapeo in Pozzuoli for 1950–2016 can be divided into three stages, each consisting of (i) early uplift and (ii) intervals of minor subsidence and recovery. Recorded VT seismicity emerged in 1971–1972 (blue columns), before decaying until 1982 (black columns) and returning to high rates in 1982–1984 (magenta columns). The numbered circles identify reference stress conditions in the crust used in Fig. 5. **(b)** The combined uplifts show a pattern of increasing VT event rates with time. **(c)** Post-uplift oscillations $\Delta h$ of 0.1–0.2 m occurred after 1952 and 1972 (1ii and 2ii), but reached 0.62 m after 1984 (3ii). The decay in VT event rate in 2ii was followed by a virtually aseismic uplift of 0.4 m during the first 8 months of renewed uplift in 1982.

stress during intrusions, interrupted by decadal intervals of approximate stasis (Fig. 2). As a result, we expect the combined episodes of uplift to show the VT-deformation behaviour of an elastic crust with a large number of small faults[23,24] (Fig. 2).

**Regimes of deformation.** The ideal sequence of behaviour starts from lithostatic equilibrium. Initial deformation is elastic, for which strain is accommodated by deformation of unbroken rock around faults (Fig. 3). As the total strain increases, the crust's behaviour becomes quasi-elastic, for which most deformation is elastic, but a small proportion is accommodated inelastically by fault movement (which is recorded as VT seismicity). The proportion of faulting increases until it becomes the only mechanism for accommodating additional strain. At this stage, the strain stored elastically remains constant and additional deformation is controlled inelastically by fault movement alone[23,24] (Fig. 3; see equations (2)–(4) in the Methods section). In addition, the rock between faults is expected to become increasingly damaged, with a greater linkage in the inelastic regime among cracks much smaller than the faults themselves[25]. The sequence finishes with bulk failure and the potential escape of magma through a newly propagating fracture. The stored strain can then be released as the crust relaxes elastically around the newly opened fracture, as well as around the pressure source[26] that caused the precursory deformation.

The quasi-elastic and inelastic regimes are described by exponential and linear trends between inelastic and total deformation[23,24] (see equations (3) and (4) in the Methods section). The total number $\Sigma N$ of VT events is a natural proxy for total inelastic deformation (not only vertical deformation), whereas the ratio $\Delta h/R$ of maximum uplift to the horizontal radius of ground uplift is a field measure proportional to total deformation. In terms of field parameters, the exponential trend for the quasi-elastic regime becomes[23,24]

$$\Sigma N = (\Sigma N)_0 \exp[(\Delta h/R)/(\lambda_{ch}/R)]$$
$$= (\Sigma N)_0 \exp(\Delta h/\lambda_{ch}) \qquad (1)$$

where $(\Sigma N)_0$ denotes the number of VT events at the start of quasi-elastic behaviour and $\lambda_{ch}$ is a characteristic displacement. Equation (1) uses the number of VT events to measure the amount of damage in the crust caused by an increase in differential stress, regardless of the source of stress.

In extension, $\Delta h/\lambda_{ch} = S_d/\sigma_T$, the ratio of differential stress to tensile strength, which has a maximum value of 4 or 5.6 for eventual bulk failure in tension or in mixed tension and shear[27–29]. Here $S_d$ refers to the accumulated differential stress in the crust after stress relaxation due to fault movement has been taken into account. Among large calderas, equation (1) has been tested[24] at Rabaul, in Papua New Guinea, where a caldera-wide uplift of 2.3 m near its centre occurred for 23 years before an intra-caldera eruption in 1994. The uplift changed from quasi-elastic to inelastic when $\Delta h/\lambda_{ch} = 4$ (Fig. 3), with the quasi-elastic regime accounting for about 80% of the total sequence[24]. Similar behaviour has been observed at stratovolcanoes, but over shorter timescales of ∼0.1–1 year. For example, the quasi-elastic regime has continued for 80% or more of total sequences with durations of several months before flank eruptions at the frequently erupting volcanoes Kilauea[23,30] and Etna[31], but for as little as 40% of the total 3-month sequences before the 2011 eruption of El Hierro in the Canary Islands[32,33], which occurred after a repose interval of more than 200 years (Fig. 4).

The repeated similarity of VT-uplift trends for different volcanoes is remarkable. It reveals a fundamental similarity in the process of damage accumulation in the crust, regardless of site-specific structures and order-of-magnitude differences in

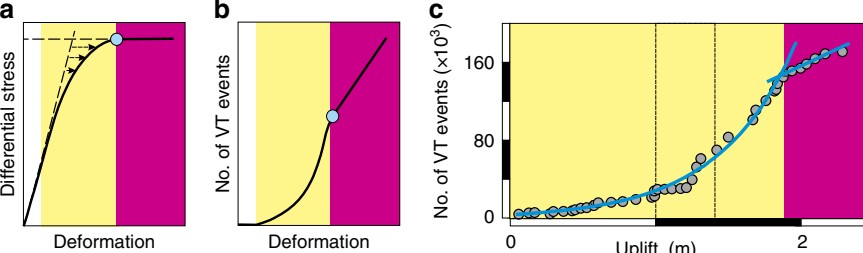

**Figure 3 | The evolution of deformation regimes during elastic-brittle behaviour.** (**a**) As the applied differential stress increases from zero, the deformation of natural rock (black curve) deviates increasingly from elastic behaviour (sloping dashed line), evolving from the elastic (white shading), through quasi-elastic (yellow) to inelastic (magenta) regimes of behaviour. The start of the inelastic regime (blue circle) coincides with deformation under a constant maintained stress. (**b**) The deviation from elastic behaviour is caused by faulting. The total deformation caused by fault movements is represented by the cumulative number of VT events. The number of VT events increases exponentially with deformation in the quasi-elastic regime, but linearly with deformation in the inelastic regime. (**c**) The evolution from quasi-elastic to inelastic deformation was observed (blue curve and line) during the 23-year precursory unrest before the 1994 eruption at Rabaul caldera, Papua New Guinea[24]. The quasi-elastic trend is given by $\Sigma N = 4{,}120 \exp(h/\lambda_{\mathrm{ch}})$, where $\lambda_{\mathrm{ch}} = 0.53$ m ($r^2 = 0.98$). The black dashed lines show the episode of rapid uplift during 1983–1985.

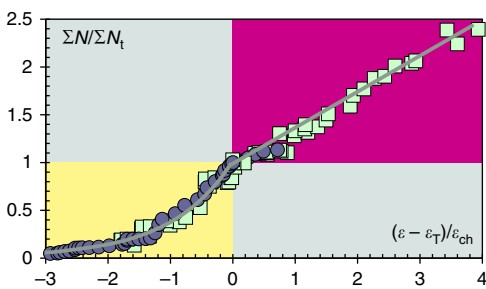

**Figure 4 | Normalized elastic-brittle trends preceding eruptions.** The VT and deformation trends have been normalized using $\Sigma N_{\mathrm{st}}$ and $\varepsilon_{\mathrm{T}}$, the number of events and strain at the end of quasi-elastic regime, and the characteristic strain $\varepsilon_{\mathrm{ch}}$. The exponential trend of the quasi-elastic regime (yellow) evolves into the linear trend for inelastic behaviour (magenta). Data are for 1971–1994 at Rabaul, Papua New Guinea (circles[24]) and for 18 July–12 October 2011 at El Hierro, Canary Islands (squares[32,33]). Field proxies for deformation were uplift at Rabaul and horizontal displacement at El Hierro (see 'Field proxies for deformation' in the Methods section.).

dimensions and process timescales, and supports our hypothesis that bulk deformation at volcanoes can be approximated to that of a crust with a large and distributed population of small discontinuities.

**Regimes of deformation at Campi Flegrei**. The combined corrected uplift at Campi Flegrei (with intervals of stasis removed) also follows the classic elastic-brittle sequence for deformation in extension (Fig. 5). The crust behaves elastically for $\Delta h < 1.75$ m and, after a short transition, becomes quasi-elastic for $\Delta h > 2.3$ m with $\lambda_{\mathrm{ch}} = 1$ m (Fig. 5). The current corrected uplift of about 4.2 m gives $\Delta h/\lambda_{\mathrm{ch}} \approx 4.2$, which suggests that the crust is now approaching the transition from quasi-elastic to inelastic deformation (Fig. 5). Virtually the same VT-uplift trend appears when using uplift uncorrected for secular subsidence (Fig. 5). Background subsidence since 1950 has thus not had a significant effect on events differential stress accumulation in the shallow crust.

The VT-uplift trend is similar to that observed at Rabaul and supports our view that the entire sequence of unrest since 1950 reflects a long-term accumulation of stress in the crust (Fig. 5). This interpretation is reinforced by the remaining interval of significant VT seismicity between 1972 and 1982 (Fig. 2), which was characterized by a gradual decay in VT event rate from 200 to

300 events per month and a minor corrected, ground subsidence and recovery of about 5% of the total uplift. This was followed by a new 30-month episode of corrected uplift that, for its first 8 months until March 1983, raised the ground at Pozzuoli by 0.4 m without significant seismicity. When VT events again occurred, they accelerated to rates of about 300–500 events per month in <3 months (Fig. 2).

The VT decay with minor ground movement resembles an extended aftershock sequence, in which fracturing and fault slip relax stresses in the surrounding rock under a constant bulk strain[34]. Before faulting can resume, the surrounding rock must be re-stressed elastically until the local stresses have returned to their values before relaxation[35]. Renewed uplift will thus occur without VT events until the stress necessary for continued faulting has been regained. From equation (1) the mean VT event rate $\mathrm{d}N/\mathrm{d}t = [\mathrm{d}(\Delta h)/\mathrm{d}t][\mathrm{d}N/\mathrm{d}(\Delta h)] = [\mathrm{d}(\Delta h)/\mathrm{d}t][\Sigma N/h_{\mathrm{ch}}]$. If the same seismic sequence is maintained across uplifts, the final VT event rate in 1972 and the starting rate in 1982 will be characterized by the same value of $\Sigma N/h_{\mathrm{ch}}$. Hence, the ratio of their respective event rates should be similar to the corresponding ratio of their mean rates of uplift. Such similarity is indeed observed: the ratio of VT event rates lies in the range $0.7 \pm 0.3$, which embraces the uplift-rate ratio of 0.8 for mean uplift rates of 0.57 m per year in 1969–1972 and 0.72 m per year in 1982–1984.

The increase in differential stress during elastic recovery is proportional to the accompanying uplift; it is also numerically equivalent to the stress previously lost by seismic relaxation. To a first approximation, stress and uplift change in proportion when behaviour is quasi-elastic[23], so that the fraction of total stress lost during relaxation is approximately the ratio of uplift during elastic recovery to total uplift before relaxation, that is 0.4 out of 2.5 m or 16%. This value is consistent with independent estimates of the proportion of energy lost by seismicity during 1972–1982. The proportion of total stress relaxed by seismicity is $\sim(E_{\mathrm{s}}/E_{\mathrm{T}})^{1/2}$, where $E_{\mathrm{s}}$ and $E_{\mathrm{T}}$ are the seismic energy released and total energy supplied[36]. Extrapolating the analysis of the 1982–1984 unrest[19,20], the seismic energy lost during 1972–1982 is $\sim 10^{13}$ J, whereas the total energy supplied until 1972 is $\sim \pi R^2 Z \rho g (\Delta h/3) \sim 10^{15}$ J, where the radius $R$ and thickness $Z$ of the deforming crust are 5 and 3 km, respectively, the mean crustal density $\rho$ is 2,200 kg m$^{-3}$, $g$ is gravity, $\Delta h$ is 2.4 m (for the interval 1950–1972) and $\Delta h/3$ is the mean uplift across the crust approximated to a cone. The estimated stress relaxation is thus $\sim (10^{13}/10^{15})^{1/2}$ or 10%.

For comparison, the seismic energy released since 1982 is c. $5 \times 10^{13}$ J (Fig. 6) or about 5% of the energy supplied during the

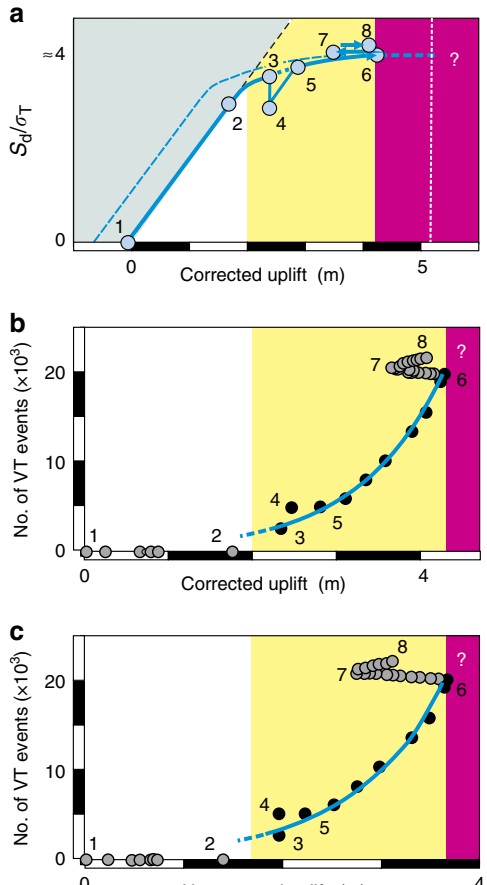

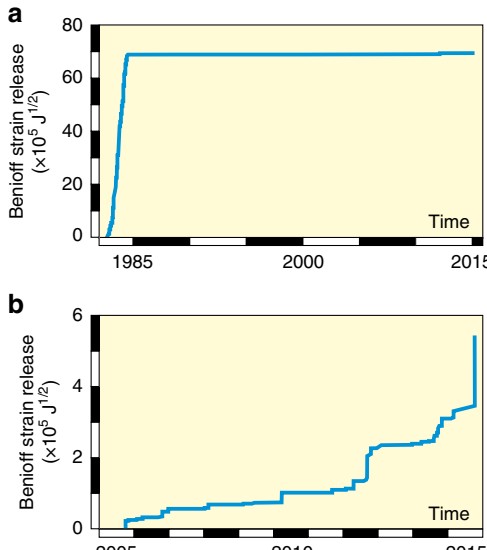

**Figure 6 | Benioff strain release at Campi Flegrei.** (**a**) Virtually all the seismicity recorded between 1982 and 2016 occurred during the 1982–1984 uplift crisis. (**b**) The slow corrected uplift of c. 0.4 m since 2005 was accompanied by a seismic energy release of $3 \times 10^{11}$ J, which approximately corresponds to 0.2% of the energy supplied during the uplift (equivalent to a relative stress drop of c. 4%). The proportion of energy lost is an order-of-magnitude smaller than the 5% estimated for the main VT-deformation trend during the 1982–1984 episode of major uplift (see the section 'Regimes of deformation in Campi Flegrei' in the main text). The slow uplift thus represents a low-seismicity return to the stress–strain condition that prevailed in 1984 (positions 6–8 in Fig. 5), which is consistent with a predominantly elastic restressing following the Kaiser effect[35].

**Figure 5 | Accumulation of stress at Campi Flegrei since 1950.** The numbers and numbered circles show reference positions in the time series in Fig. 2. (**a**) The trend (solid blue curve) shows an overall accumulation of stress and increase in the proportion of inelastic deformation, interrupted by partial stress relaxation (3–4) and recovery (4–5) during 1972–1982. Ground oscillation since 1984 is consistent with a drop (6–7) to lower pore pressure (blue dashed curve) and subsequent recovery (7–8). Deformation occurs in the elastic (white shading) and quasi-elastic (yellow) regimes, and approaches the inelastic (magenta) regime. (**b**) The variation of total number of VT events with combined corrected uplift during the rapid uplifts of 1950–1952, 1969–1972 and 1982–1983 (1i + 2i + 3i in Fig. 2) shows a short transition from elastic to quasi-elastic behaviour. In the quasi-elastic regime, the VT event number increases as $\Sigma N = 295 \exp(h/\lambda_{ch})$ and $\lambda_{ch} = 1$ m ($r^2 = 0.99$). A return to the main VT-uplift trend (blue solid curve) may coincide with the emergence of inelastic crustal deformation. (**c**) The variation of total number of VT events with uncorrected uplift shows almost the same trend as for corrected uplift, with $\Sigma N = 408 \exp(h/\lambda_{ch})$ and $\lambda_{ch} = 1$ m ($r^2 = 0.98$).

additional increase in $\Delta h$ by c. 1.8 m. The corresponding reduction in stress is c. 20%. The estimated proportion of seismic stress release has thus increased with time since the onset of unrest in 1950. Nevertheless, some 80% of the stress applied has remained accumulated in the crust and is the amount represented by $S_d$. The result confirms that the uplift at Campi Flegrei to date has been determined primarily by elastic deformation, rather than by fault movement.

Unlike the 1969–1972 unrest, the uplifts of 1950–1952 and 1982–1984 were not followed by decays in VT event rate. This is expected for the earlier episode of elastic deformation. The 1982–1984 sequence, in contrast, continued the quasi-elastic response that had been established in 1969–1972 but, instead of a

decreasing VT event rate, uplift was followed by an abrupt cessation in seismicity and, in 33 years with fewer than 2,000 VT events, a corrected subsidence of c. 0.62 m by 2000 and its almost complete recovery by 2017 (Figs 2 and 6).

Corrected subsidence without seismicity is favoured by a contemporaneous decrease in either or both the differential stress applied to the crust and the pore-fluid pressure within the crust. Differential stress is generated by magma overpressure, which can be decreased by reducing the volume of magma by gas loss on vesiculation or by thermal contraction on solidification. At Campi Flegrei, the magmatic sills causing each episode of unrest have thicknesses of metres. These solidify within years[15] and so are not able to accommodate movements over 16 years. Reductions in differential stress through magmatic action are thus unlikely controls on the corrected subsidence since 1984. The corrected movement, however, can be accommodated by the relaxation of pore pressure in the geothermal system by the diffusion of pressurized fluids[9,16,19,37]. Diffusion is suggested also for the uplift since 2000, because both uplift and subsidence have occurred at similar rates and lengths of time, with variations in the influx of magmatic fluids from depth being a preferred control on the geothermally driven ground movement[37–42].

Viewed as a single sequence, therefore, unrest at Campi Flegrei can be explained by the evolving deformation of an elastic-brittle shallow crust. This first quantitative interpretation of the caldera's long-term behaviour shows that there is no need to require significant non-brittle flow due to viscous[19] or plastic[43] movements at timescales of ~10 years. Thus, during the 1969–1972 uplift, the bulk behaviour evolved from elastic to quasi-elastic and may now be close to the next transition from quasi-elastic to inelastic. The VT-uplift trend, in particular, is following that observed at Rabaul before its 1994 eruption and

suggests that long-term stress accumulation may be a general feature of unrest at large calderas.

## Discussion

Our interpretation predicts that if the current uplift continues to a corrected value of about 4.5 m at Pozzuoli, the crust in Campi Flegrei will have returned to the stress conditions that prevailed in 1984 at the end of the last major uplift (Fig. 6). We would then expect any additional uplift to continue the VT-deformation trend interrupted in 1984 and, hence, to be accompanied by a significant increase in VT seismicity, regardless of the specific mechanism that is increasing the applied differential stress. Should the rate of uplift also return to the rapid values of 1982–1984, we would further expect the onset of VT event rates as high as 800–1,000 per month. Rapid uplift, however, is not essential. At Rabaul, for example, the approach to eruption was preceded by 2 years at a maximum recorded uplift rate of about 0.15 m per year, which was about three times smaller than the peak rates that had been registered 10 years previously[24]. A return to the long-term VT-deformation trend at Campi Flegrei may thus occur at uplift rates and VT event rates slower than observed during previous emergencies.

The indirect stress ratio $\Delta h/\lambda_{ch}$ suggests that the differential stress accumulated in Campi Flegrei's crust is about four times its tensile strength (Fig. 5) and so is approaching the transition from quasi-elastic to inelastic deformation regimes. An increase in linkage among small-scale cracks between faults is also expected to occur at the transition to inelastic behaviour. This would favour an increase in bulk permeability and, hence, a faster escape of fluids from the geothermal system, which is consistent with the onset of corrected subsidence in 1984. A return to the long-term VT-deformation trend may therefore be characterized by inelastic behaviour under a constant maintained stress, for which increases in total deformation are determined by additional fault movement (Fig. 3). Such a transition would be associated with VT event rates increasing in proportion to the rate of uplift.

The few field data available for large calderas and stratovolcanoes suggest that the quasi-elastic regime contributes between 40 and 80% of the total precursory deformation (Fig. 4). Assuming this range, a corrected uplift of 4.2 m at the end of quasi-elastic behaviour at Campi Flegrei (Fig. 5) indicates that the inelastic regime may continue until reaching a total corrected uplift of between 5 and 10 m before an eruption can be expected. A transitional value of 4 for $\Delta h/\lambda_{ch}$ assumes that bulk failure occurs in tension. The value increases towards 5.6 as the failure mechanism involves tension with an increasing component of shear[27–29]. Increasing shear could thus raise the transitional uplift by some 25% and, hence, yield a total corrected uplift of between 6.25 and 12.5 m before an eruption.

The estimated limits on total uplift are smaller than the 17 m of caldera-wide uplift inferred to have occurred during the century before the caldera's last eruption in 1538 (refs 2,6). A greater total uplift would be favoured by a larger uplift before the transition to inelastic behaviour, without necessarily changing the proportion of uplift in the two deformation regimes, or by a greater proportion of uplift in the inelastic regime alone. A larger transitional uplift would be favoured if the pre-1538 intrusions had been required to break connected horizons of rock stronger than those providing resistance today (to increase the uplift required before tensile failure). Otherwise, the difference may indicate that mechanisms for reducing effective bulk rigidity, such as bedding-plane slip[43,44], become significant as deformation proceeds (to enable greater uplift for a given applied stress); that, at timescales of $\sim 10^2$ years, non-brittle (and seismically quiet) processes, such as viscous flow[19], also

contribute to deformation (to permit greater uplift than from elastic-brittle behaviour); that additional intervals of fault slip under constant strain reduce the accumulated stress (to enable a greater total uplift before the failure stress is eventually achieved); or that fluid pressure in the hydrothermal system has become large enough to contribute significant uplift.

Although these mechanisms would favour a greater proportion of inelastic deformation at Campi Flegrei than has been recorded elsewhere, none of them guarantees that an uplift of 17 m needs to occur before eruption. The onset of inelastic behaviour thus represents a significant increase in the potential for volcanic activity and provides a new criterion for defining levels of alert. In common with other volcanoes for which few or no precursory data are available from previous eruptions[45–47], expert elicitation is a favoured method for evaluating unrest at Campi Flegrei[48]. The method estimates the probability of an eruption given the occurrence of selected precursory criteria, such as critical rates or amounts of ground uplift. By necessity, the critical values are determined empirically from volcanoes elsewhere and so are not well constrained[49]. However, the VT-deformation trends (Fig. 3) are generic and can be applied in the absence of historical information. The change from the quasi-elastic to inelastic regime therefore complements probabilistic evaluations by providing an objective criterion for increasing alert levels.

At Campi Flegrei itself, an additional obstacle to effective warning is a low public awareness of volcanic hazard compared with the perceived threat from microseismicity[50,51]. The persistent VT seismicity in 1983–1984 damaged buildings throughout Pozzuoli and triggered the evacuation of some 40,000 people[52]. Compared with emergencies since 1950, therefore, a new episode of rapid uplift is likely to present a greater hazard from persistent ground shaking, as well as a significant increase in the potential for eruption. Past experience of rapid uplifts is thus unreliable for perceiving the level of risk during a future emergency. The residents of Campi Flegrei have experienced three episodes of rapid uplift over seven decades without an eruption. This favours the view that rapid uplifts are poor indicators of imminent volcanic activity. Recognizing the long-term evolution in precursory behaviour is essential for moderating misplaced confidence in non-eruptive outcomes and for delivering improved warnings to the public.

## Methods

**Quantifying regimes of elastic-brittle deformation.** The VT event rate is controlled by stresses around the peripheries of faults, where damage zones develop with dimensions much smaller than the faults themselves[23]. The mean differential stress across damage zones $S_{dz} = S_d + S_{tf}$, where $S_d$ is the net applied differential stress and $S_{tf}$ is the mean difference between the stress gained by transfer from adjacent crust relaxing during faulting and the stress lost by creating and opening discontinuities in the damage zones. Increases in $S_{dz}$ are thus limited by increases in either $S_d$ or in $S_{tf}$, corresponding to rates of faulting limited by increases in bulk stress or in local stress transfer. By inspection, therefore, quasi-elastic deformation is associated with bulk-stress faulting and inelastic deformation with stress-transfer faulting.

From thermodynamics, the probability that damage zones fracture is given by $\exp[-(S_{st} - S_{dz})/S_{ch}]$, where $S_{st}$ is mean rock strength, $S_{st} - S_{dz}$ is the additional stress required for bulk fracture and the characteristic stress $S_{ch}$ is the maximum equivalent stress available from stochastic fluctuations in atomic configuration. The mean rate of inelastic deformation with supplied differential stress, $d\varepsilon_{in}/dS_{sup}$, is then[23]

$$d\varepsilon_{in}/dS_{sup} = (d\varepsilon_{in}/dS_{sup})_{af}\exp[(S_d + S_{tf} - S_{st})/S_{ch}] \qquad (2)$$

where the attempt frequency $(d\varepsilon_{in}/dS_{sup})_{af}$ is the frequency with which the stochastic fluctuations in stress attempt to break the damage zones. $S_{sup}$ is the differential stress supplied before taking account of stress drops due to fault movement, whereas $S_d$ is the maintained stress after the seismic stress drops have been removed. The value for $S_{ch}$ depends on the style of deformation. Failure in compression is limited by shearing between atoms, but in extension by the tearing of bonds. As reflected by macroscopic properties, $S_{ch}$ in compression depends on temperature and effective confining pressure, for which $S_{ch} \equiv S^* = (3\Phi T + P_c - P_p)/3$, where $T$ is absolute temperature (K), $P_c$ and $P_p$ are the confining and pore-fluid

pressures, and $\Phi$ is the molecular energy per unit volume per temperature[23]. In extension $S_{ch}$ defines the tensile strength $\sigma_T$ of unbroken rock and is effectively constant for the pressures and temperatures in the crust beneath volcanoes.

Equation (2) shows that the rate of inelastic deformation with stress depends on the difference between $S_d + S_{tf}$ and $S_{st}$. Initial implementations[53] of the model considered the limiting condition for which the stress difference is controlled by a reduction in $S_{st}$, through processes such as chemically enhanced stress corrosion. These were subsequently generalized[23] to conditions for which the rate of stress drop by faulting balanced the rate of applied stress increase without the need to invoke chemical rock weakening; in this case, the rate of inelastic deformation is determined by increases in $S_{tf}$.

In the quasi-elastic limit, $S_d \approx S_{sup} = Y\varepsilon$, where $Y$ is Young's modulus, and $S_{tf}$ is negligible. Assuming that the stress distribution about the mean is constant and that the total number, $\Sigma N$, of VT events is proportional to inelastic strain ($\Sigma N = C\, d\varepsilon_{in}$), integration of equation (2) yields

$$\Sigma N = \Sigma N_{st}\exp[(\varepsilon - \varepsilon_{st})/\varepsilon_{ch}] \qquad (3)$$

where $\varepsilon_{st} = S_{st}/Y$, $\varepsilon_{ch} = S_{ch}/Y$ and $\Sigma N_{st}$ is the number of VT events when $S_d \approx S_{st}$ at the start of the inelastic regime. In this example, the failure strain $\varepsilon_{st}$ is assumed to be constant, which implies that any weaking processes affect both failure strength and Young's modulus in the same proportion.

In the inelastic limit, $S_d$ is held approximately constant, because the mean rate of stress drop by faulting balances the mean rate of stress supplied by the pressure source. Additional increases in total strain are controlled by inelastic deformation alone ($d\varepsilon_{in}/d\varepsilon \approx 1$), for which

$$\Sigma N_{in} = \Sigma N_{in}, 0 + C(\varepsilon_{in} - \varepsilon_{in,0}) = \Sigma N_{in}, 0 + C(\varepsilon - \varepsilon_{in,0}) \qquad (4)$$

where $\Sigma N_{in,0}$ ($\geq \Sigma N_{st}$) is the number of VT events before the start of the inelastic regime.

**Field proxies for bulk deformation.** Assuming a constant geometry of deformation, common field measures of bulk strain include ground tilt, uplift and horizontal displacement. The preferred choice depends on the form of monitoring network and on which parameter yields the largest variation. Maximum uplift $\Delta h$ is the chosen parameter at Campi Flegrei, so that $\Delta h = K\varepsilon$ and $\lambda_{ch} = K\varepsilon_{ch}$, where $K$ is a constant of proportionality. With these substitutions, equation (3) yields equation (1) in the main text.

**Data availability.** All relevant data are available from the authors.

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

## Acknowledgements

We are grateful to Giovanna Berrino (INGV-OV) for providing geodetic databases, Alexander Steele (UCL) for database integration and Danielle Charlton (UCL) for cartographic design. We also thank Agust Gudmundsson and an anonymous reviewer for constructive comments that improved our initial manuscript.

## Author contributions

C.R.J.K. developed the elastic-brittle model and wrote the manuscript. All authors contributed to the interpretation of seismic and geodetic data.

## Additional information

**Competing interests:** Part of the work by C.R.J.K. was funded through the CITYVOLC Project, sponsored by Aon Benfield Reinsurers (www.aonbenfield.com). The remaining authors declare no competing financial interests.

