## [Peer Review File · Nature Communications]

Reviewers' Comments:

Reviewer #1 (Remarks to the Author)

Review of the manuscript entitled "Progressive approach to eruption at Campi Flegrei caldera, southern Italy" by Christopher Kilburn, Giuseppe De Natale, and Stefano Carlino, submitted for publication to Nature Communications

This manuscript applies a new model of elastic brittle failure to the Campi Flegrei caldera, to test the idea that successive episodes promote a long-term accumulation of stress in the crust, thus providing the first quantitative evidence that this volcanic system is evolving towards conditions more favourable to eruption and identifying field tests for predictions on how the caldera will behave during future unrest. The authors oppose this new view to the classical idea that individual episodes of unrest represent independent events, so that only data from an ongoing episode are considered pertinent to evaluating eruptive potential, thus leading to the implicit assumption that the crust relaxes accumulated stress after each episode.

The paper is well written and presents an interesting approach to a problem that, certainly, is still far from being fully understood. About the novelty of the approach proposed it is true that, as far as I know, this is the first time that it is applied to Campi Flegrei, particularly for the consideration of the seismic catalogue (N value), but similar approaches using RSAM to measure the tensile fracturing rate (in this paper the parameter $\Delta h/\lambda_{ch}$) have been used in other active volcanic areas.

Caldera systems are very complex, structurally speaking. These are highly fractured systems in which determination of the proper rheology of the rocks is not simple. This implies that models like the one presented here are just approximations that may result rather simplistic, although the idea they rely on should be taken into account in further attempts to understand what is currently going on in Campi Flegrei. Particularly, I like the new view that all unrests post-dating the last eruption occurred in this caldera should be considered as steps of the same process rather than as individual events. However, I find some limitations not commented by the authors that I consider important when dealing with caldera systems:

1) Δh , is the variable that measures inelastic deformation and it is equated to the deformation at the vertical component (up-lift), not taking into account the lateral movements. Also, the authors put together this information with the number of earthquakes at the start of the quasi-elastic behaviour, to make the final number equal to the total number of VT events. For this assignment to be correct, all the earthquakes should have a main vertical movement with the same mechanism than the fault, as they sum the displacement of all earthquakes and equal the final result to the total uplift. However, this is not necessarily correct in a caldera system where different types of movements, even reverse, may account for the seismic information recorded. Information on focal mechanisms should be provided here in order to support the assumption made by the authors

2) Seismicity associated with tensile fracturing is of very small magnitude and more in the case of a caldera system, which is assumed to be already highly fractured. The lowest magnitudes will be unnoticed and the recorded seismic events, which will constitute the seismic catalog, will mostly correspond to the response of the whole volcanic edifice to the deformation, which will try to assimilate the local stress changes in a similar way as in an induced seismicity. Therefore, it must be ensured that N measures the number of earthquakes responsible for the mechanism that forces the surface uplift to reach a value Δh . In the case of a caldera system, the existence of numerous stress barriers created by structural discontinuities, presence of rheological contrasts, etc, may be

significant in the control of seismicity induced by overpressures from the magmatic or hydrothermal systems, so it must be ensured that it is possible to separate the seismicity induced by pressure changes from that corresponding to structural readjustments of the host rock. This also needs clarification here

3) The comparison with the Rabaul caldera I am not sure should provide similar numbers. The internal structure and dynamics of both calderas are not so similar and the proof is that unrest episodes in both of them do not follow similar patterns. So, I would not use a predictive assumption for Campi Flegrei based on what was observed in Rabaul

Also, I assume that Nature readers cover a much wider audience than specialised journals. In some of the concepts used in this paper (eg: volcano-tectonic, VT, event; quasi-elastic and inelastic regimes, etc) should be defined somewhere to ensure that potential readers will fully understand the paper.

Reviewer #2 (Remarks to the Author)

This paper is on a very important and interesting caldera, namely Campi Flegrei in Italy. The caldera is important from a hazard point of view because it has given rise to some of the largest explosive eruptions in Europe in the past tens of thousands of years. It is also important for the same reason because of great uplift (doming) which has occurred in the past decades, and which the authors explain through shallow sill emplacement. Presently, some millions of people live close to, or inside, the caldera. I make the following comments:

1. The authors propose the idea that the stress (and strain) is accumulating in Campi Flegrei in the past decades (cumulative uplift or doming of the order of 3-4 metres). They suggest that this concentration of stress will eventually result in eruption, but explain that this is still far less uplift than the one (about 17 m) associated with the last main eruption. I think the authors' ideas are very plausible and valid. They might add some sentences to the effect that it is really the strain energy accumulation that will then largely control the resulting eruption and possible caldera collapse at the time of magma chamber rupture and eruption. Thus the greater the strain energy accumulated in the volcano before the next eruption, the larger is the available energy to squeeze magma out the chamber during the eruption and contribute to the caldera collapse. This has been discussed in several papers (e.g. Strengths and strain energies of volcanic edifices: implications for eruptions, collapse calderas, and landslides, nness, 2012) and should be mentioned even if the authors do not have space to cite the above paper.

2. The authors mention the earthquakes during the unrest periods. I did not see, however, if they were able to estimate the stress release (or strain energy release) through the cumulative effects of the seismicity. This should be easy to do, if they have not done so already, and compare with the strain energy/stress concentration generated in the volcano in the uplift periods.

3. The ms is clearly written. All the illustrations are needed and well made.

The topic is of the greatest interest and importance and deserves to be published for a wide audience, hence its suitability for Nature Communications. I recommend acceptance after minor revision that takes these comments into account.

Response to Reviewers:

Reviewer #1

Comment. I like the new view that all unrests post-dating the last eruption occurred in this caldera should be considered as steps of the same process rather than as individual events. However, I find some limitations not commented by the authors that I consider important when dealing with caldera systems.

Δh , is the variable that measures inelastic deformation and it is equaled to the deformation at the vertical component (uplift), not taking into account the lateral movements. Also, the authors put together this information with the number of earthquakes at the start of the quasi-elastic behaviour, to make the final number equal to the total number of VT events. For this assignation to be correct, all the earthquakes should have a main vertical movement with the same mechanism than the fault, as they sum the displacement of all earthquakes and equal the final result to the total uplift. However, this is not necessarily correct in a caldera system where different types of movements, even reverse, may account for the seismic information recorded. Information on focal mechanisms should be provided here in order to support the assumption made by the authors.

Response. Δh is a proxy for total deformation and *not* only for vertical deformation. Similarly, ΣN is a proxy for total inelastic deformation. This has been emphasized in Lines 117-120.

We have declared that most of the VT events have involved normal fault movements (Lines 95-96).

We start counting the number of earthquakes when they are detected. No assumption has been made *a priori* about quasi-elastic behaviour. To the contrary, the data themselves reveal a quasi-elastic trend.

Comment. Seismicity associated with tensile fracturing is of very small magnitude and more in the case of a caldera system, which is assumed to be already highly fractured. The lowest magnitudes will be unnoticed and the recorded seismic events, which will constitute the seismic catalog, will mostly correspond to the response of the whole volcanic edifice to the deformation, which will try to assimilate the local stress changes in a similar way as in an induced seismicity. Therefore, it must be ensured that N measures the number of earthquakes responsible for the mechanism that forces the surface uplift to reach a value Δh . In the case of a caldera system, the existence of numerous stress barriers created by structural discontinuities, presence of rheological contrasts, etc, may be significant in the control of seismicity induced by overpressures from the magmatic or hydrothermal systems, so it must be ensured that it is possible to separate the seismicity induced by pressure changes from that corresponding to structural readjustments of the host rock. This also needs clarification here

Response. The relation between Δh and total deformation has been addressed above. We are measuring damage in the crust due to differential stress, regardless of the source of that stress (Line 125).

Field data show the two VT-deformation regimes among precursors to eruptions in numerous tectonic settings and among volcanoes that erupt frequently and after long repose. The data are described in Lines 129-138 and Figures 4 & 5, which demonstrate that the approach is not an over-simplification. We suggest that the data instead show that conventional approaches are over-complicated. We have also described the derivation of the elastic-brittle model in a new Methods Section, rather than providing only citations to previous studies.

The role of the hydrothermal system at Campi Flegrei is distinguished specifically in Lines 200-211 and Figure 6.

Comment. The comparison with the Rabaul caldera I am not sure should provide similar numbers. The internal structure and dynamics of both calderas are not so similar and the proof is that unrest episodes in both of them do not follow similar patterns. So, I would not use a predictive assumption for Campi Flegrei based on what was observed in Rabaul.

Response. We have described analogies with other volcanoes, in addition to Rabaul, in Lines 129-138 and 240-244, as well as Figure 5.

Comment. I assume that Nature readers cover a much wider audience than specialised journals. In sense some of the concepts used in this paper (eg: volcano-tectonic, VT, event; quasi-elastic and inelastic regimes, etc) should be defined somewhere to ensure that potential readers will fully understand the paper.

Response. VT events are defined in Lines 91-92 and the nature of the deformation regimes in Lines 106-112, with additional specialist details in the Methods Section.

Reviewer #2

Comment. The authors might add some sentences to the effect that it is really the strain energy accumulation that will then largely control the resulting eruption and possible caldera collapse at the time of magma chamber rupture and eruption. Thus the greater the strain energy accumulated in the volcano before the next eruption, the larger is the available energy to squeeze magma out the chamber during the eruption and contribute to the caldera collapse. This has been discussed in several papers (e.g. Strengths and strain energies of volcanic edifices: implications for eruptions, collapse calderas, and landslides, nness, 2012) and should be mentioned even if the authors do not have space to cite the above paper.

Response. The magnitude of an eruption is beyond the scope of this paper. However, we have included the potential role of elastic strain-energy release in driving magma to the surface in Line 115, as well as the recommended reference (*Ref. 25*).

Comment. The authors mention the earthquakes during the unrest periods. I did not see, however, if they were able to estimate the stress release (or strain energy release) through the cumulative effects of the seismicity. This should be easy to do, if they have not done so already, and compare with the strain energy/stress concentration generated in the volcano in the uplift periods.

Response. Estimates of the seismic energy released and its relation to changes in accumulated stress have been included in Lines 178-193, as well as Figure 7.

Reviewers' Comments:

Reviewer #1 (Remarks to the Author)

This revised version of the manuscript entitled "Progressive approach to eruption at Campi Flegrei caldera, southern Italy" by Christopher Kilburn, Giuseppe De Natale, and Stefano Carlino, submitted for publication to Nature Communications, has been significantly improved with respect to the original version. It is now much clearer and concise, and has corrected the drawbacks contained in the original version. New information added to the supplementary material is more than welcome and helps to understand some critical aspects of the manuscript. My comments and suggestions have all been considered and well addressed in this revision, as well as the others made by a second referee. As I mentioned in my first review, the model of elastic brittle failure applied to the Campi Flegrei caldera, assuming that successive episodes promote a long-term accumulation of stress in the crust, is an interesting idea to provide direct field evidence that this volcanic system is evolving towards conditions more favourable to eruption. I agree with the authors that this offers an alternative to a more subjective experts elicitation method, which provides probabilistic approaches based on selected precursory criteria. The method proposed by Kilburn et al is based on field data and rock mechanics, so it offers a more tangible way to evaluate the proximity of an eruption at Campi Flegrei, by predicting how the caldera may behave in future unrests. I think that this is a novel and good contribution to science, so I recommend the publication of this manuscript in Nature Communications.

Reviewer #2 (Remarks to the Author)

This paper is on a very interesting and important topic, namely the precursors to eruptions, with application to Campi Flegrei, Italy, perhaps the most dangerous volcano in Europe. Following earlier reviews the authors have significantly modified their manuscript. They have taken into account my comments as well as those of the other reviewer, and I feel the manuscript is now much better and more clearly written than it was before.

I think the modifications are as great as can fairly be expected. Also, I think the material presented has very wide interests - particularly because Campi Flegrei regularly enters unrest periods with great uplifts and earthquake swarms, and partly because this is perhaps, as said before, the dangerous volcano in Europe.

Given the changes and the wide interest of the topic, I recommend acceptance.

Agust Gudmundsson